# Monocular Object-Level SLAM Enhanced by Joint Semantic Segmentation and Depth Estimation

**DOI:** 10.3390/s25072110

**Published:** 2025-03-27

**Authors:** Ruicheng Gao, Yue Qi

**Affiliations:** 1State Key Laboratory of Virtual Reality Technology and Systems, School of Computer Science and Engineering, Beihang University, Beijing 100191, China; 2Qingdao Research Institute of Beihang University, Qingdao 266104, China

**Keywords:** object-level SLAM, semantic segmentation, depth estimation, multi-task learning

## Abstract

SLAM is regarded as a fundamental task in mobile robots and AR, implementing localization and mapping in certain circumstances. However, with only RGB images as input, monocular SLAM systems suffer problems of scale ambiguity and tracking difficulty in dynamic scenes. Moreover, high-level semantic information can always contribute to the SLAM process due to its similarity to human vision. Addressing these problems, we propose a monocular object-level SLAM system enhanced by real-time joint depth estimation and semantic segmentation. The multi-task network, called JSDNet, is designed to predict depth and semantic segmentation simultaneously, with four contributions that include depth discretization, feature fusion, a weight-learned loss function, and semantic consistency optimization. Specifically, feature fusion facilitates the sharing of features between the two tasks, while semantic consistency aims to guarantee the semantic segmentation and depth consistency among various views. Based on the results of JSDNet, we design an object-level system that combines both pixel-level and object-level semantics with traditional tracking, mapping, and optimization processes. In addition, a scale recovery process is also integrated into the system to evaluate the truth scale. Experimental results on NYU depth v2 demonstrate state-of-the-art depth estimation and considerable segmentation precision under real-time performance, while the trajectory accuracy on TUM RGB-D shows less errors compared with other SLAM systems.

## 1. Introduction

Simultaneous localization and mapping (SLAM) determines the 6-DoF pose of a moving camera and establishes a map of the unknown scene, which enables a variety of applications in augmented reality (AR)/virtual reality (VR)/ mixed reality (MR), robotics, and autonomous driving [1,2,3,4,5,6,7,8]. Compared with stereo- or RGB-D-based techniques, monocular SLAM algorithms are attractive to many mobile applications in indoor scenarios because of cheap hardware, a simple calibration process, and no limitations in the depth range.

However, with only monocular RGB input, two kinds of constraints appear. Firstly, the compound of scale drift and accumulative error may incapacitate data collection entirely. Secondly, the hand-crafted point features are always sensitive to environmental changes such as light or dynamic factors. By contrast, for humans scale drift does not occur routinely even if acting as cyclopean observers. Moreover, human vision systems can fuse various kinds of prior information. For example, from a single image, people can basically detect objects, perceive spatial layout, and estimate approximate depth.

To avoid the above limitations, semantic information from deep networks is exploited by different strategies, covering object detection, semantic segmentation, and saliency detection [9,10,11,12,13]. On the one side, with prior semantic inputs, both localization accuracy and robustness are enhanced due to the higher stability of high-level semantic information, which is also more similar to human vision. On the other side, through the typical size of objects, semantic information is always helpful to recover the truth scale in monocular SLAM [14]. However, the intrinsic error in object size leads to less accuracy than in RGB-D SLAM systems. The estimated depth result of the deep network also contributes to the scale recovery, but the long tail may reduce the accuracy of depth prediction. Meanwhile, with the development of multi-task deep networks, research on joint learning between depth prediction and semantic segmentation indicates that the two tasks can improve each other [15,16].

Inspired by the above, we enhance the monocular SLAM system with real-time joint depth estimation and semantic segmentation, aiming to solve the scale drift and improve the localization performance, as shown in Figure 1. Firstly, a real-time joint network called JSDNet is proposed to achieve simultaneous depth prediction and semantic segmentation. Secondly, with the evaluated results, we design an object-level semantic SLAM system based on ORB-SLAM2 [17]. The proposed system enhances the common SLAM process covering feature matching, local mapping, and global bundle adjustment (BA), implementing object-level SLAM. Furthermore, scale optimization is added to recover the scale with depth prediction. In summary, our work makes the following contributions.
1.We propose JSDNet for joint learning of depth estimation and semantic segmentation, focusing on fixed depth estimation and a feature fusion block.2.We suggest a semantic consistency process to keep the spatial consistency of semantic segmentation and depth estimation, with the aim of not only contributing to the two tasks, but also improving localization robustness.3.We design an object-level SLAM system based on JSDNet with the utilization of pixel-level and object-level semantic information. In detail, the system optimizes the process covering feature matching, and local and global BA to improve accuracy and robustness. In addition to this, we add a scale uniformization procedure to recover a stable scale.

To validate the performance of the above contributions, experiments on both depth and segmentation datasets demonstrate the considerable results. Moreover, results on TUM RGB-D show accurate localization performance.

In the following, related works are summarized in Section 2. Section 3 illustrates the architecture of JSDNet. The design of the SLAM system is introduced in Section 4. Section 5 reports the experimental results of JSDNet and object-level SLAM system. Finally, Section 6 draws conclusions.

## 2. Related Work

### 2.1. Monocular Visual SLAM

In recent years, keyframe-based SLAM systems have come to predominate, such as DTAM [18], and ORB-SLAM [19] and its extended systems [17,20]. Monocular SLAM methods have been developed for accurate, robust, and efficient tracking and mapping. With the development of SLAM, keyframe-based and direct methods are two main kinds of techniques. Keyframe-based SLAM approaches represent the map by a few selected frames and enhance the localization performance through BA optimization. Among these methods, the representative system PTAM [21] processes tracking and mapping tasks in parallel threads to estimate motion and builds the map of the unknown scene. DTAM [18] minimizes a global energy function to produce a surface patchwork with millions of vertices, with hundreds of images as inputs instead of a simple photometric data term. ORB-SLAM2 [17] utilizes fast ORB features and bags of words for tracking, relocalization, loop closure, and mapping, achieving state-of-the-art pose estimation accuracy. ORB-SLAM3 [20] exploits maximum a posteriori estimation during the IMU initialization phase to improve localization robustness. DM-VIO [22] uses delayed marginalization and pose graph bundle adjustment into visual–inertial odometry. DynaVINS [23] uses a robust bundle adjustment that can reject the features from dynamic objects by leveraging pose priors estimated by IMU preintegration.

### 2.2. Semantic SLAM

Semantic SLAM combines semantic information with the traditional pipeline to obtain more accurate and robust performance [24,25,26,27]. DynaSLAM [24] adds dynamic object detection and background inpainting by multi-view geometry, deep learning, or both. Moreover, DynaSLAM II [26] optimizes jointly static and dynamic parts with the trajectories of both the camera and the moving agents using a novel bundle adjustment proposal. DS-SLAM [25] combines a deep network with a moving consistency check method to improve robustness in dynamic environments. Frost et al. [14] adopted semantic information to recover the ground truth scale by adding scale evaluation into the BA process. CubeSLAM [12] proposes both single-image detection and multi-view object SLAM, and demonstrates that the two parts can improve each other. RigidFusion [28] presents a novel RGB-D SLAM approach to simultaneously segment, track, and reconstruct the static background and large dynamic rigid objects. YOLO-SLAM [29] proposes a dynamic-environment-robust visual SLAM system with a lightweight object detection network and a new geometric constraint method. Qiu et al. [30] proposed a novel method to resolve the object-scale ambiguity with a generic image-based two-dimensional tracker, which enables accurate metric three-dimensional tracking of arbitrary objects. Dynam-SLAM [31] loosely couples the visual scene stream with an IMU for dynamic feature detection and then optimizes the data measured using tight coupling. DGM-VINS [32] exploits a joint geometric dynamic feature extraction module and a temporal instance segmentation module to enhance system stability and localization accuracy. RLD-SLAM [33] combines object detection and Bayesian filtering to maintain high accuracy while quickly acquiring static feature points.

Different from these ideas, we enhance the performance of monocular SLAM with joint segmentation and depth estimation, exploiting pixel-level and object-level semantic information.

### 2.3. Joint Semantic Segmentation and Depth Estimation

As the depth and semantic labels share context information, many researchers focus on joint depth estimation and semantic segmentation [16,34,35,36]. Silberman et al. [37] proposed an RGB-D dataset with color, depth, and semantic information called NYU. Then, an extended dataset called SUN RGB-D with about ten thousand training and testing images was proposed [38]. Based on this dataset, Eigen et al. [39] addressed depth prediction, surface normal estimation, and semantic labeling with a multi-scale convolutional network. Furthermore, more CNN models [40,41] were proposed to improve segmentation performance through leveraging depth information. On the contrary, prior semantic labels can also improve depth prediction. Jiao et al. [15] designed a synergy network with an attention-driven loss to adopt semantic information to improve depth estimation. Focusing on time consumption, Nekrasov et al. [16] proposed a real-time network with asymmetric annotations for these two tasks. CI-Net [36] presented a network injected with contextual information to improve the accuracy of the two tasks.

The network presented in [16] satisfies the real-time requirements for SLAM systems. However, the accuracy of both branches still needs to be improved, especially for the depth estimation. To solve these limitations, we transform the depth to segmentation labels, focusing on the fixed range instead of the full depth. Moreover, we propose a fusion block, semantic consistency, and corresponding loss function to make the two branches improve each other.

## 3. JSDNet

### 3.1. Architecture

Improvements of JSDNet. Compared with previous works, four improvements are made to improve depth and segmentation precision in JSDNet. Firstly, we convert the depth regression to segmentation by a discretization strategy, addressing a fixed range instead of the full range. Secondly, to make the two branches improve each other, we define a feature fusion block in the decoder part. Thirdly, the learning-weighted loss function is designed to balance the two branches. Finally, we use the semantic consistency strategy to further improve depth and segmentation accuracy.

Depth discretization. In terms of depth prediction, JSDNet outputs depth information with a fixed depth range to achieve more accurate results. The method for this goal is to transform the depth regression task to a classification task with a spacing-increasing discretization strategy. The detailed formula is as follows:(1)$$\begin{array}{cc} \; & DS\left(w,h\right)=\left\{\begin{array}{c} 0,\;\;\;D(w,h)<=\alpha ,\\ \frac{\ln(D(w,h)-\alpha +1.0)}{\ln(\beta -\alpha +1.0)})\times \left(K-2\right),\\ (\alpha <=D(w,h)<=\beta ),\\ K-1,\;\;D(w,h)>=\beta , \end{array}\right. \end{array}$$
where α and β denote the minimum and maximum values of the depth range, *K* represents the truncated number, and D(w,h) is the original depth value.

Feature fusion. We propose a feature fusion block in JSDNet to make the two branches improve each other. As demonstrated in Figure 2, JSDNet is composed of three branches covering feature extraction, depth estimation, and segmentation. Taking real-time property as the main consideration, JSDNet selects MobileNet_v3 [42] as the backbone. In the decoder part, both depth prediction and segmentation adopt a mirror-network structure using multi-scale features. The two branches keep the same architecture, composed of four blocks with each block containing two convolution layers: one CRB layer [16] and one upsample layer.

In particular, before the last three CRB layers, JSDNet sums the original outputs and fused features to implement parameter sharing between the two decoders. The architecture of the feature fusion block is shown in the purple box of Figure 2. Firstly, the depth and semantic features are applied with a convolution layer, and then concatenated. Secondly, the fused features are added to the original features, and a convolution layer is subsequently employed. Thirdly, the above two steps are performed n times to obtain the final outputs. In the actual implementation, we set n=2. Obviously, the fusion block and the depth and semantic branches can affect each other, which the aim of improving the accuracy of both tasks, as shown in later experiments.

### 3.2. Loss Function

Corresponding to the outputs, the loss function also includes the segmentation and depth estimation terms. Since JSDNet converts depth regression to segmentation, both semantic and depth branches use the cross-entropy loss. The purpose of the depth estimation branch is to recover the ground truth scale value using Formula (Equation 10). As a consequence, the depth loss multiplies an inversion of the depth value by the cross-entropy loss. To balance these two losses, JSDNet uses homoscedastic uncertainty [43] with the following formula:(2)$$\begin{array}{cc} L= & exp\left(-D\right)\;\cdot \;L_{d}+D+exp\left(-S\right)\;\cdot \;L_{s}+S, \end{array}$$(3)$$L_{s}=-\frac{1}{W*H}\sum_{w=0}^{W}\sum_{h=0}^{H}\sum_{i=0}^{L}Y_{i}^{g}\left(w,h\right)*\log\left(Y_{i}^{p}\left(w,h\right)\right),$$(4)$$\begin{array}{cc} L_{d}= & -\frac{1}{W*H}[\sum_{w=0}^{W}\sum_{h=0}^{H}[\sum_{i=0}^{K}{\mathrm{DS}}_{i}^{g}\left(w,h\right)\\ \; & *\log({\mathrm{DS}}_{i}^{p}\left(w,h\right))]/D\left(w,h\right)], \end{array}$$
where *W* and *H* denote the image’s width and height, $Y_{i}^{p}$ and $Y_{i}^{g}$ represent the prediction and ground truth of the semantic labels, ${\mathrm{DS}}_{i}^{p}$ and ${\mathrm{DS}}_{i}^{g}$ express the prediction and ground truth of the depth labels, and *D* and *S* are parameters learned by the network.

### 3.3. Semantic Consistency

Semantic consistency process. To keep the consistency of depth estimation and segmentation, we design a consistency optimization process as follows. Firstly, we select any two frames (denoted as $F_{1},F_{2}$) of images in the same scene, and obtain the depth and segmentation from JSDNet. Secondly, based on the camera pose of $F_{1},F_{2}$, we transform the depth and semantic labels of $F_{1}$ into the camera space of $F_{2}$. Then, the loss (Equation 2) is employed to train JSDNet. Thirdly, similar to the second step, we also calculate the loss by transforming the camera space of $F_{2}$ to $F_{1}$.

Semantic consistency dataset. With the above step, JSDNet can be trained with spatial consistency. Moreover, it is noticeable that the dataset for training semantic consistency does not need to contain semantic and depth labels. Instead, the dataset only includes RGB and its corresponding camera pose, which is easy to obtain.

Semantic consistency training. In particular, in each round of training, JSDNet is firstly trained by the ground truth depth and semantics, and then trained by consistency optimization.

## 4. Semantic SLAM

### 4.1. Visual Odometry

Traditional point features are always sensitive to environmental changes such as lighting or dynamics. We leverage semantic and depth information to enhance feature matching in visual odometry in terms of three aspects. Firstly, we add semantic label matching to the original ORB feature matching process. Secondly, the pixel matching in textureless areas and dynamic areas such as ground, wall, window, and person is removed. Thirdly, we discard the matching that goes out of the depth scope.

### 4.2. Three-Dimensional Object Representation and Generation

Compared with point feature-based maps, object-level representations are always more stable. In our SLAM system, we adopt both the 3D rectangular bounding box and hand-crafted feature points to represent the 3D object. For convenience, the bounding box of the 3D object is parameterized as 6 degrees covering the position and object size. The orientation is set the same as the three coordinate axes.

The construction process of semantic objects is the following. Through the combination of semantic segmentation and depth estimation, the 3D bounding box under the camera coordinate system is generated. Then, by means of the obtained camera pose with feature matching, the 3D object is mapped into the world space. Furthermore, 3D objects in the world space from different frames are judged in terms of uniqueness according to the union of objects and point feature matching.

By the above process, we establish the map with semantic objects, combining low-level point features with high-level semantic features. The built map is later exploited in the optimization process to achieve more accurate camera poses.

### 4.3. Object-Level Bundle Adjustment

By means of the traditional BA process, multiple components such as map points and camera poses are jointly optimized. In our implementations, the object constraint is also added to the optimization procedure, taking advantage of high-level semantic information to enhance accuracy and robustness. Camera poses, points, and objects are denoted as $C=C_{i}$, $P=P_{i}$, $O=O_{i}$, respectively, for convenience, then BA can be expressed as the following nonlinear optimization problem:(5)$$\begin{array}{cc} C^{*},P^{*},O^{*}= & argmin\sum_{C_{i},P_{i},O_{i}}(∥E\left(P_{i},C_{i}\right)∥\\ \; & +∥E\left(P_{i},O_{i}\right)∥+∥E\left(O_{i},C_{i}\right)∥), \end{array}$$
where $E\left(P_{i},C_{i}\right),E\left(P_{i},O_{i}\right),E\left(O_{i},C_{i}\right)$ represent the camera-point, object-point and camera-object errors.

In the aspect of the camera-point error, we adopt the standard 3D point reprojection error from ORB-SLAM [17,19], with the following form:(6)$$\begin{array}{c} E\left(P_{i},C_{i}\right)=\pi \left(T_{c}P_{i}\right)-z, \end{array}$$
where z denotes the pixel coordinates of the 3D point, $T_{c}$ and π represent the transformation matrices from world to camera and from camera to pixel. The object-point error measures the spatial relations between feature points and 3D objects. If a pixel point is identified as belonging to an object, then the corresponding 3D point in world coordinates should be located exactly on the object’s surface. This property can be utilized to design the object-point error by comparing the distance between the feature point and the center of the 3D object in pixel coordinates, which can be computed in two ways, either through a series of projections or measured directly in the pixel space:(7)$$\begin{array}{c} E\left(P_{i},O_{i}\right)=∥\pi \left(T_{c}\left(P_{i}-O_{i}\left(c\right)\right)\right)-\left(O_{i}\left(s\right)-c\right)∥, \end{array}$$
where $O_{i}\left(c\right)$ and $O_{i}\left(s\right)$ denote the center and size of the object, respectively, and c represents the center of the object in pixel coordinates.

As Figure 3 demonstrates, each object in the world space is reprojected onto the image plane. Then, the object-camera error is calculated by comparing the reprojected 2D bounding box with the detected 2D bounding box from JSDNet.(8)$$\begin{array}{cc} E(O_{i},C_{i}) & =∥\pi \left(T_{c}O_{i}\left(c\right)\right)-O_{2d}\left(c\right)∥\\ \; & +∥\pi \left(T_{c}O_{i}\left(s\right)\right)-O_{2d}\left(s\right)∥. \end{array}$$

### 4.4. Scale Restoring

After visual odometry and back-end optimization with additional semantic and depth information, the camera pose is refined and the semantic map is established. However, the scale drift and ambiguity problems still exist. In this subsection, we design a scale recovery process that outputs a scale-unified map with estimated depth information.

As Figure 1 shows, the scale recovery process is added to the system after local mapping. The problem is defined as follows:(9)$$\begin{array}{c} s=argmin\sum_{p\in \mathrm{MP}}\sum_{k\in \mathrm{Obs}(p)}{∥s\times d\left(T_{c}\left(k\right)p\right)-d_{p}∥}_{2}, \end{array}$$
where MP represents the set of map points, Obs(p) denotes the observation keyframes of point p, and $d_{p}$ is the estimated depth value. For convenience, $d(T_{c}\left(k\right)p)$ is marked as $d_{g}$. Then, the solution to (Equation 9) is obtained as follows:(10)$$\begin{array}{c} s=\frac{\sum_{p\in \mathrm{MP}}\sum_{k\in \mathrm{Obs}(p)}d_{p}d_{g}}{\sum_{p\in \mathrm{MP}}\sum_{k\in \mathrm{Obs}(p)}d_{p}^{2}}. \end{array}$$

Through the above formula, the scale value is estimated.

## 5. Experiments

### 5.1. Datasets and Implementation Details

Datasets. To verify the performance of our framework, we conduct experiments on NYU depth v2 [37] with depth estimation and segmentation, and on TUM RGB-D [44] with camera localization.

NYU depth v2 [37] is an indoor dataset with 795 training images and 654 testing images. The dataset is comprised of video sequences from a variety of indoor scenes as recorded by both the RGB and depth cameras from the Microsoft Kinect, which can support joint depth estimation and semantic segmentation.

TUM RGB-D [44] comprises a large number of color and depth images captured using a Kinect sensor. The ground truth camera pose of each image is estimated through a high-accuracy motion capture system. We select eight sequences from the dynamic object category to verify the dynamic localization performance.

Implementation details. In JSDNet, the RGB image is unified to 480×480 pixels and enhanced with scale, flip, and padding. The depth information is truncated into 120 labels from 0.5 m to 3 m. In addition, two extra labels are added to denote depths of less than 0.5 m and more than 3 m. The feature extraction of JSDNet adopts the MobileNet_v2 architecture pretrained on the ImageNet dataset. To balance the depth and segmentation loss, learned coefficients of camera pose *S* and *D* are both initialized as 0.0. Moreover, the training strategy in the work [16] is also adopted. In addition to the above, JSDNet leverages the ADAM optimizer with an initial learning rate of $8\times 10^{-3}$ for MobileNet_v3 and $8\times 10^{-4}$ for the decoder. The implementation of our SLAM system is based on ORB-SLAM2. In the match process, the feature pixels with depths of more than 3.0 m or less than 1.0 m are discarded. Meanwhile, the textureless and dynamic semantic labels like wall, ceiling, window, and person are also removed from the match.

In the object generation process, objects identified with fewer than 10 feature points are also discarded. Moreover, objects generated from different frames are considered identical if the IoU is larger than 0.6 and the feature points match. In the BA process, the objects observed by more than 60% of keyframes are added to the optimization.

### 5.2. Depth and Segmentation Results

Depth estimation results. In terms of depth prediction, we exploit the root mean square error (RMSE), average relative error (Rel), root mean square error in log space (RMSE-Log), and accuracy with threshold (δ) as metrics, which is the same as previous works [16,34,35,36]. In detail, the δ metric is expressed by $max\left(\frac{d_{g}}{d_{p}},\frac{d_{p}}{d_{g}}\right),\delta_{1}=1.25,\delta_{2}=1.25^{2},\delta_{3}=1.25^{3}$, where $d_{g}$ and $d_{p}$ denote the depth ground truth and prediction, respectively.

Table 1 lists the depth estimation results on NYU depth v2 compared with multi-task methods, including those of Eigen et al. [39], Mousavian et al. [34], Nekrasov et al. [16], SOSD-Net [35], CINet [36] Hoyer et al. [45], and Lopes et al. [46]. All methods focus on joint depth estimation and semantic segmentation.

For the absolute metrics (RMSE and RMSE-Log), it is obvious that our result with depths of 0.5 m–3 m obtains the least error. Moreover, with full depth estimation, we also obtain a lower error compared with other methods. In the aspect of the relative metrics (Rel, $\delta_{1}$, $\delta_{2}$, and $\delta_{3}$), we also achieve the most accurate results. In short, both the above results demonstrate the state-of-the-art performance of our method.

Semantic segmentation results. The semantic segmentation results are illustrated in Table 2 on NYU depth v2 with pixel accuracy (Pixel-acc), mean accuracy (Mean-acc), and mean intersection over union (IoU) as metrics. The contrasted approaches are the same as for depth prediction in Table 1. It is obvious that our results are more accurate on Pixel-acc and IoU, but less accurate than SOSD-Net [35] on Mean-acc.

Running time. In terms of inference speed, JSDNet achieves 12.1 milliseconds for one image with a size 640×480 on an Nvidia 3090 GPU, satisfying the real-time need of SLAM systems. Compared with the real-time approach [16], we obtain a 4.9% improvement in the IoU metric.

Visualization results. The top part of Figure 4 states the visualization results of depth and segmentation on TUM RGB-D. It is noticeable that TUM RGB-D does not contain the training semantic information. The results are obtained by training JSDNet on the whole of SUN RGB-D.

### 5.3. Object-Level SLAM Result

Static scenes. The quantitative results of localization accuracy on TUM RGB-D are shown in Table 3. In static scenes with monocular input, it is clearly seen that our SLAM system obtains less camera trajectory error compared with ORB-SLAM2 [17] and Luo et al. [9]. The localization improvements are 82.9%,69.2%,45.7%,81.4%, respectively, in the four experimental sequences compared with ORB-SLAM2, proving the effects of the estimated depth and semantic segmentation. When the input covers the additional depth information, our system also achieves better performance than ORB-SLAM2. The improvements show that the utilization of semantic information contributes to the more accurate localization results.

Dynamic scenes. In dynamic scenes, we compare with ORB-SLAM2 [17], DS-SLAM [25], DynaSLAM [24], Refusion [47], LC-CRF SLAM [48], DGM-VINS [32], Wang et al. [49], and SG-SLAM [50] in Table 3. It is obvious that our SLAM system with RGB-D input achieves the best performance. Moreover, lines 1, 2, and 9 demonstrate that the monocular system outperforms ORB-SLAM2 and DS-SLAM. This is due to the depth estimation improvement with the fixed range.

Camera pose visualization. The bottom part of Figure 4 states the camera pose visualization results predicted by our SLAM result in both static (fr2_xyz, fr2_office) and dynamic scenes (fr3_walking_xyz). It is obvious that the pose sequences remain stable, which demonstrates localization robustness.

### 5.4. Ablation Studies

Discussion of outstanding performance. In the above experiments, the results demonstrate state-of-the-art performance on depth estimation, semantic segmentation, and camera localization. In our opinion, the reason for improvement lies in three aspects. Firstly, we design the feature fusion module for feature sharing jointly using depth estimation and semantic segmentation. Secondly, we propose the semantic consistency process to guarantee depth and semantics in different views, which not only improves the two tasks, but also contributes to feature matching in camera localization. Thirdly, we put forward object-level BA for camera pose and map optimization that combines hand-crafted features and predicted semantic object features. In the following, we conduct ablation experiments to prove the effect of these modules.

Ablation studies of feature fusion block and semantic consistency on depth estimation and semantic segmentation. To validate the effects of the feature fusion block and semantic consistency, we perform ablation studies on both segmentation and depth estimation, with the results shown in Table 4. In the settings S2 and S3 of Table 4, we remove feature fusion and semantic consistency, respectively. It is clear that both the RMSE and IoU metrics decrease, which proves the positive effect of the two modules. Moreover, we remove both the modules in the setting S1. It can be seen that the accuracy further decreases compared with S2 and S3.

Ablation studies of feature fusion, semantic consistency, and object-level BA on localization. In S1–S8 of Table 5, we show the results of ablation studies on the impact of the three modules on localization. All the results in the table are obtained by using RGB-D as input. When there is no object-level BA process, we use the traditional BA defined in ORB-SLAM2 instead. Without feature fusion and semantic consistency, we also exploit the predicted depth and segmentation from the visual odometer.

In settings S5,S6 and S7, we remove feature fusion, semantic consistency, and object-level BA, respectively. It can be see that the localization errors all increase compared with S8, proving the positive effect of the three modules. Moreover, the localization error order of these four settings is S5(0.030m)>S7(0.028m)>S6(0.017m)>S8(0.016m). The results indicate that the feature fusion module has a small effect on localization (0.016m to 0.017m), while the effects of semantic consistency and object-level BA are large (0.016m to 0.028m and 0.030m). The explanations are as follows. The feature fusion module mainly aims to improve depth estimation and segmentation, but this has little effect on localization. In contrast, semantic consistency not only improves the two tasks, but also contributes to feature matching in the visual odometer, hence generates a large effect on the localization results. In settings S2,S3 and S4, two modules are removed. From the result pairs S3 and S7 and S4 and S5, the accuracy increase with feature fusion is further validated. Meanwhile, improvements with semantic consistency and object-level BA can also be proven. When we remove the three modules (S1), the localization generates the most increase, which sufficiently proves the positive effect of the three modules on localization.

## 6. Conclusions

In this paper, paying attention to localization robustness and accuracy of the monocular SLAM system, we study the object-level SLAM system design with the enhancement of joint depth estimation and semantic segmentation. We first propose a joint learning network called JSDNet that outputs both depth estimation and semantic segmentation. To improve the performance of JSDNet, a fusion block and semantic consistency modules are designed. Specifically, the feature fusion combines depth estimation features with segmentation features, with the aim of improving both tasks, while semantic consistency guarantees depth and segmentation consistency under different views. To train JSDNet, we design an appropriate loss function with learned weights to balance the two tasks. Based on the results of JSDNet, we define the object-level BA error with additional camera-object and object-point constraints to improve localization accuracy and robustness. Additionally, we design a scale recovery procedure to obtain a stable scale. To validate the performance, experiments of JSDNet and the proposed SLAM system are conducted. First, in terms of depth estimation and semantic segmentation on NYU depth v2, we achieve state-of-the-art performance compared with other methods while keeping real-time performance. Second, our object-level SLAM system obtains a lower RMSE error on both static and dynamic scenes of the TUM RGB-D dataset. Moreover, ablation experiments of the feature fusion, semantic consistency, and object-level BA are conducted, which sufficiently prove the positive effect of the three modules on depth prediction, semantic segmentation, and localization.

In this paper, we make a good attempt at embedding the predicted depth and segmentation into the SLAM system with visual odometer and optimization. For future work, it is recommended to investigate a method that combines semantics with multi-source data, such as RGB, depth, inertial measurement units, and so on. In our opinion, this may further improve the localization robustness.

## Figures and Tables

**Figure 1 sensors-25-02110-f001:**
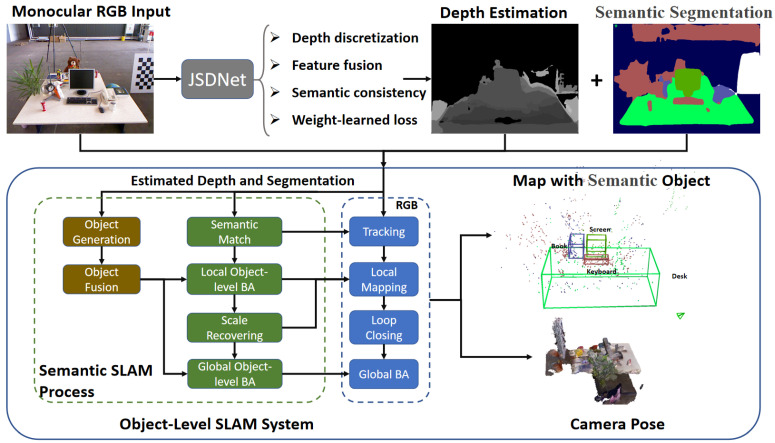
Outline of whole SLAM system. With estimated depth and segmentation results from JSDNet, the object-level SLAM system is implemented with enhancement of tracking, local mapping, and global BA processes.

**Figure 2 sensors-25-02110-f002:**
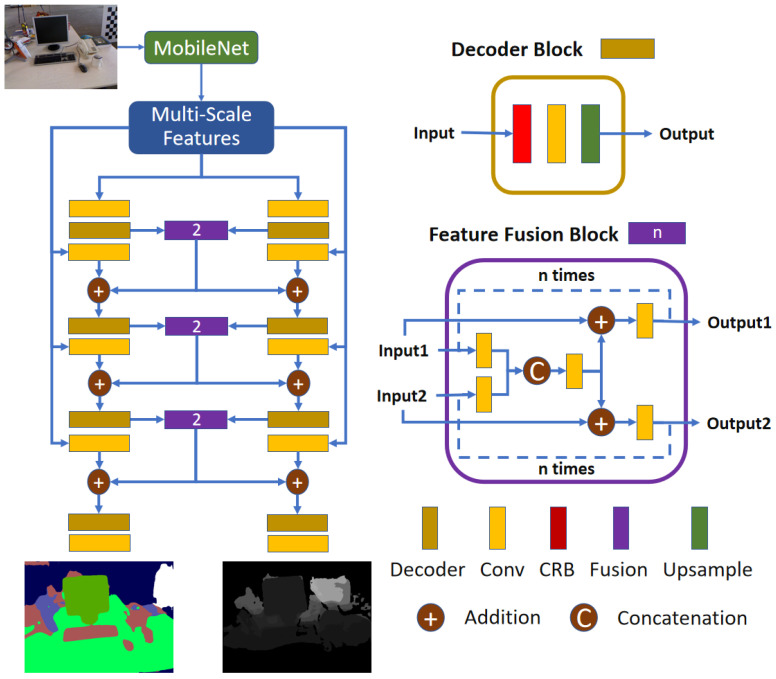
Architecture of JSDNet. JSDNet regresses depth and segmentation with a monocular RGB image as input.

**Figure 3 sensors-25-02110-f003:**
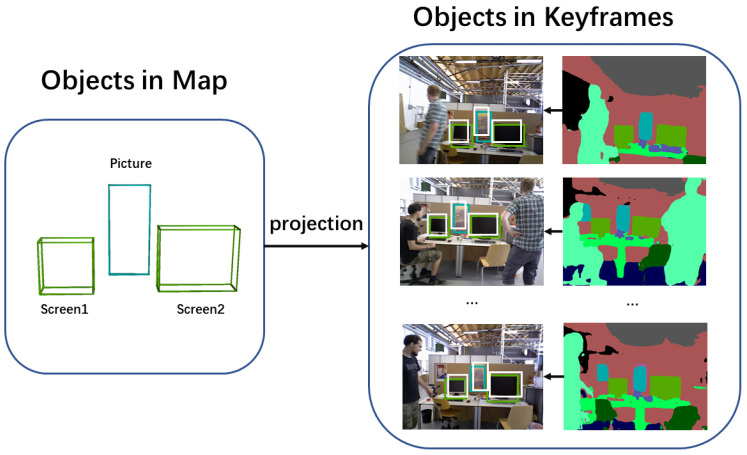
Reprojection of semantic objects in map onto the keyframe plane. White boxes state the reprojection of objects in map, while green and blue boxes denote results detected by semantic segmentation.

**Figure 4 sensors-25-02110-f004:**
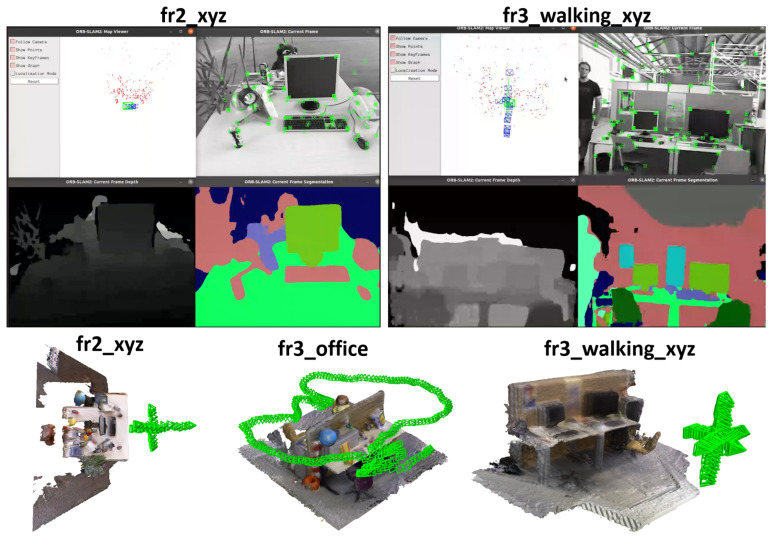
Visualization of depth estimation, semantic segmentation, and predicted camera poses on TUM RGB-D dataset.

**Table 1 sensors-25-02110-t001:** Depth estimation results compared with other methods on NYU depth v2.

Method	RMSE	Rel	Log	$\delta_{1}$	$\delta_{2}$	$\delta_{3}$	Speed
Eigen et al. [39]	0.641	0.158	0.214	0.769	0.950	0.988	-
Mousavian et al. [34]	0.816	0.200	0.314	0.568	0.856	0.956	-
Nekrasov et al. [16]	0.565	0.149	0.205	0.790	0.955	0.990	12.6
SOSD-Net [35]	0.514	0.145	-	0.805	0.962	0.992	-
CI-Net [36]	0.504	0.129	0.181	0.812	0.957	0.990	-
Hoyer et al. [45]	0.622	-	-	-	-	-	-
Lopes et al. [46]	0.604	-	-	-	-	-	-
Ours (0.5 m–3.0 m, no segmentation)	0.358	0.148	0.148	0.811	0.961	0.988	-
Ours (Full depth, no segmentation)	0.543	0.148	0.192	0.816	0.959	0.992	-
Ours (0.5 m–3.0 m)	**0.335**	0.138	**0.132**	0.821	0.968	0.994	-
Ours (Full depth)	0.495	**0.128**	0.168	**0.838**	**0.992**	**0.995**	**12.1**

**Table 2 sensors-25-02110-t002:** Semantic segmentation results compared with other methods on NYU depth v2.

Method	Pixel-Acc	Mean-Acc	IoU
Eigen et al. [39]	65.6	45.1	34.1
Mousavian et al. [34]	68.6	52.3	39.2
Nekrasov et al. [16]	-	-	42.0
SOSD-Net [35]	72.2	**62.5**	43.3
CI-Net [36]	72.7	-	42.6
Hoyer et al. [45]	-	-	39.5
Lopes et al. [46]	-	-	38.9
Ours (no depth estimation)	69.8	53.2	40.7
Ours	**76.4**	61.1	**46.9**

**Table 3 sensors-25-02110-t003:** RMSE of absolute trajectory error (m) results on TUM RGB-D dataset.

**Static Scenes (Monocular)**					
	fr1_xyz	fr2_xyz	fr1_desk	fr3_office	Mean
ORB-SLAM2	0.041	0.039	0.035	0.865	0.245
Luo et al. [9]	0.027	0.026	0.192	0.635	0.22
Ours	**0.007**	**0.012**	**0.019**	**0.161**	**0.050**
**Static Scenes (RGB-D)**					
	fr1_xyz	fr2_xyz	fr1_desk	fr3_office	Mean
ORB-SLAM2	0.005	**0.004**	0.016	0.010	0.009
Ours	**0.003**	**0.004**	**0.011**	**0.008**	**0.007**
**Dynamic Scenes (fr3_walking)**					
	xyz	static	rpy	half	Mean
ORB-SLAM2 [17] (RGB-D)	0.459	0.090	0.662	0.351	0.391
DS-SLAM [25] (RGB-D)	0.025	0.008	0.444	0.030	0.127
DynaSLAM [24] (RGB-D)	0.015	**0.006**	0.035	0.025	0.020
ReFusion [47] (RGB-D)	0.099	0.017	-	0.104	-
LC-CRF SLAM [48](RGB-D)	0.016	0.011	0.046	0.028	0.025
DGM-VINS [32] (RGB-D)	0.036	0.013	0.071	0.033	0.038
Wang et al. [49] (RGB-D)	0.015	0.007	0.047	0.023	0.023
SG-SLAM [50] (RGB-D)	0.015	0.007	0.032	0.020	
Ours (Monocular)	0.039	0.010	0.076	0.045	0.043
Ours (RGB-D)	**0.014**	**0.006**	**0.028**	**0.017**	**0.016**

**Table 4 sensors-25-02110-t004:** Ablation studies on the effects of feature fusion and semantic consistency on depth estimation (RMSE) and segmentation (IoU).

	Feature Fusion	Semantic Consistency	RMSE	IoU
S1			0.517	43.8
S2	*√*		0.505	45.3
S3		*√*	0.508	45.5
S4	*√*	*√*	0.495	46.9

**Table 5 sensors-25-02110-t005:** Ablation studies on the effects of feature fusion, semantic consistency, and object-level BA on localization performance.

	Feature Fusion	Semantic Consistency	Object-Level BA	xyz	Static	rpy	Half	Mean
S1				0.044	0.021	0.082	0.049	0.049
S2	*√*			0.037	0.019	0.077	0.044	0.044
S3		*√*		0.033	0.016	0.064	0.039	0.038
S4			*√*	0.029	0.010	0.061	0.033	0.033
S5	*√*		*√*	0.028	0.011	0.052	0.030	0.030
S6		*√*	*√*	0.015	0.007	0.030	0.017	0.017
S7	*√*	*√*		0.024	0.010	0.049	0.029	0.028
S8	*√*	*√*	*√*	0.014	0.006	0.028	0.017	0.016

## Data Availability

The original data presented in the study are openly available in [37,44].

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
