# Peer review of "Monocular Object-Level SLAM Enhanced by Joint Semantic Segmentation and Depth Estimation"

_sensors, 2025, doi:10.3390/s25072110_

Round 1

Reviewer 1 Report

Comments and Suggestions for Authors

Summary of the Paper

This paper presents a semantic SLAM system that incorporates object-level information. Specifically, a neural network, JSDNet, predicts both depth and semantic masks for each input monocular image. The object-level information is then integrated into a monocular SLAM pipeline (ORB-SLAM2) with several modifications:

  1. Both traditional low-level feature matching and high-level semantic matching are considered.

  2. An object-level bundle adjustment (BA) is introduced under the assumption that the projection of an object's 3D bounding box should be close to its 2D bounding box projection.

  3. Scale is recovered based on the predicted depth.

The proposed method is validated on real datasets, and the results demonstrate its effectiveness.

Overall, this is a solid paper for Sensors

Questions

  • What is the generalizability of the designed JSDNet?

  • Given that an object's geometry may vary across different viewing directions, I am somewhat concerned about whether the projection of the 3D bounding box’s center is consistently close to that of the 2D bounding box.

The paper is generally well-written and easy to follow.

Comments on the Quality of English Language

The paper is generally well-written and easy to follow.

Reviewer 2 Report

Comments and Suggestions for Authors

The author proposes a multi-task network named JSDNet, which performs depth estimation and semantic segmentation at the same time, and integrates semantic features for localization.
1. I am not sure whether the JSDNet proposed by the author is based on the improvement of the existing network model. If so, it is recommended to quote relevant references and explain the differences.
2. is the JLDNet mentioned in line 173 of the paper correct?
3. Does JSDNet have the ability to combat the impact of noise data on depth estimation and semantic segmentation results?

Comments on the Quality of English Language

Suggestions for Improvement Consistency: Ensure consistent terminology throughout the paper. For example, “object-level” should consistently be hyphenated. Clarity: Some sentences could be rephrased for better clarity and flow. For example, “the feature fusion implements feature sharing between the two tasks” could be rephrased as “feature fusion facilitates the sharing of features between the two tasks.” Grammar: Pay attention to subject-verb agreement and the correct use of articles. For example, “the scale recovering process” should be “the scale recovery process.” 
